# A Blockchain-Based Secure Data Transaction and Privacy Preservation Scheme in IoT System

**DOI:** 10.3390/s25154854

**Published:** 2025-08-07

**Authors:** Jing Wu, Zeteng Bian, Hongmin Gao, Yuzhe Wang

**Affiliations:** 1China Mobile Information Technology Co., Ltd., Beijing 100876, China; 2Beijing University of Posts and Telecommunications—China Mobile Communications Group Co., Ltd. Joint Institute, Beijing 100876, China; 3School of Cyberspace Security, Beijing University of Posts and Telecommunications, Beijing 100876, China

**Keywords:** IoT, blockchain, data trading, privacy protection, attribute-based encryption, homomorphic encryption

## Abstract

With the explosive growth of Internet of Things (IoT) devices, massive amounts of heterogeneous data are continuously generated. However, IoT data transactions and sharing face multiple challenges such as limited device resources, untrustworthy network environment, highly sensitive user privacy, and serious data silos. How to achieve fine-grained access control and privacy protection for massive devices while ensuring secure and reliable data circulation has become a key issue that needs to be urgently addressed in the current IoT field. To address the above challenges, this paper proposes a blockchain-based data transaction and privacy protection framework. First, the framework builds a multi-layer security architecture that integrates blockchain and IPFS and adapts to the “end–edge–cloud” collaborative characteristics of IoT. Secondly, a data sharing mechanism that takes into account both access control and interest balance is designed. On the one hand, the mechanism uses attribute-based encryption (ABE) technology to achieve dynamic and fine-grained access control for massive heterogeneous IoT devices; on the other hand, it introduces a game theory-driven dynamic pricing model to effectively balance the interests of both data supply and demand. Finally, in response to the needs of confidential analysis of IoT data, a secure computing scheme based on CKKS fully homomorphic encryption is proposed, which supports efficient statistical analysis of encrypted sensor data without leaking privacy. Security analysis and experimental results show that this scheme is secure under standard cryptographic assumptions and can effectively resist common attacks in the IoT environment. Prototype system testing verifies the functional completeness and performance feasibility of the scheme, providing a complete and effective technical solution to address the challenges of data integrity, verifiable transactions, and fine-grained access control, while mitigating the reliance on a trusted central authority in IoT data sharing.

## 1. Introduction

With the rapid development of Internet of Things (IoT) technology, massive amounts of IoT data are being generated at an unprecedented rate, becoming the core driving force for global economic growth and industrial transformation. As IoT data becomes an important asset in modern society, its status in the digital economy continues to improve. As a key link in promoting digital transformation and intelligent upgrading, data trading and circulation have become the core driving force for promoting social and economic development. However, with the rapid growth of IoT data output and the continuous growth of market demand, how to ensure the efficiency of these data circulation while effectively protecting privacy has become an urgent problem to be solved. In data trading, privacy protection faces unprecedented challenges. Studies [1] have shown that the process of data trading inevitably involves the circulation of a large amount of personal privacy information, which needs to be properly protected; otherwise, it will pose a threat to personal privacy and data security.

In this context, privacy computing technology has attracted widespread attention as a solution that can make data “available but invisible” [2]. This technology allows data processing and analysis without exposing sensitive information, and enables data to be calculated in an encrypted state, thereby effectively protecting user privacy while ensuring data processing. This is particularly important for data transactions, as it allows both parties to the transaction to verify the validity of the data without revealing the data content.

The traditional data circulation model [1] struggles to meet the needs of large-scale and widespread applications under the existing regulatory framework and data protection mechanism. With the rapid development of blockchain technology, more and more industries are beginning to try to apply blockchain to data transaction circulation. The introduction of blockchain technology provides a new guarantee for data circulation. The decentralized, tamper-proof and traceable characteristics of blockchain make it an important technology to ensure the transparency and credibility of data transactions. However, while ensuring the immutability of data, blockchain has also raised concerns about privacy protection [3,4]. Since blockchain is a public ledger, all transaction records will be made public, so how to circulate data without leaking sensitive information has become a research focus. Although blockchain technology can effectively prevent data tampering, there are still loopholes in data privacy, especially when it comes to personal sensitive information. How to ensure data privacy through encryption technology or zero-knowledge proof has become an important research direction [3]. On the basis of ensuring the transparency and credibility of blockchain transactions, how to properly protect the privacy information has become a difficult problem that urgently needs to be solved. Taking Bitcoin and other digital currencies as an example, although blockchain technology has the advantages of low cost and easy circulation, its privacy protection problem has always been one of the key factors restricting its development. Users generally do not want their sensitive information to be disclosed on the blockchain, and this unrestricted information exposure may bring unforeseen security risks [4].

Therefore, this paper will explore the following core research question: How to design a comprehensive IoT data ecosystem framework to enable secure, fine-grained data transactions and privacy-preserving computations without relying on a centralized trusted third party? Specifically, it aims to address three key sub-problems: (1) establishing a decentralized access control mechanism for heterogeneous IoT devices; (2) creating a fair and dynamic data transaction pricing model; and (3) performing complex analytical computations on encrypted data to protect user privacy throughout the data lifecycle. To specifically validate and illustrate the effectiveness of this framework, this article examines the Internet of Medical Things (IoMT) as a primary application scenario. As a key branch of the Internet of Things (IoT), data in the IoMT field is highly sensitive and valuable, and imposes stringent security and privacy requirements. Therefore, the detailed mechanisms and scenarios analyzed in subsequent chapters will focus on the IoMT.

The rest of this paper is organized as follows: Section 2 reviews the related work on privacy protection and data sharing based on blockchain. Section 3 introduces the preliminary cryptographic techniques used in the scheme, including blockchain, homomorphic encryption, and attribute-based encryption. Section 4 introduces the detailed design of the proposed framework, including system architecture and algorithms. Section 5 conducts a comprehensive security analysis of the scheme. Section 6 describes the experimental setup and evaluates the performance of the prototype system. Finally, Section 7 concludes this paper and discusses future work.

## 2. Related Work

This chapter aims to review existing work related to data privacy protection and secure transactions in the Internet of Things based on blockchain. The content will be reviewed from three aspects: privacy protection schemes, data transaction sharing schemes, and data stream computing schemes, in order to clarify the current status, challenges, and positioning of this research.

### 2.1. Privacy Protection Scheme

With the popularity of IoT devices, a large amount of highly sensitive data such as personal behavior, location, and health is continuously collected, making privacy protection a key issue. The traditional centralized data management model that relies on a single trusted entity is no longer suitable for today’s environment due to its inherent single point of failure, separation of data ownership and control, and susceptibility to data tampering and abuse. While it must be acknowledged that cutting-edge cryptographic techniques such as encoded computation [5] and verifiably encoded computation [6] can provide strong levels of privacy and even information-theoretic security, they still typically operate in systems that assume the existence of a central coordinator. These approaches mainly solve the problem of “how to compute securely”, but fail to solve the fundamental problems of centralized trust, data sovereignty, and auditability brought by the architecture itself.

For example, while the work of Zyskind et al. [7] laid the foundation for blockchain access control, its design failed to fully consider the resource-constrained nature of IoT devices, which is exactly the gap that this work aims to fill. Recent research has further promoted the development of this field, especially in optimizing system overhead, enhancing data privacy protection and secure sharing. For example, to address the challenge of limited resources in IoT devices, Mohanta et al. [8] proposed an edge computing solution that integrates blockchain, AES encryption and off-chain IPFS storage. By storing a large amount of data off-chain, the load of the blockchain is significantly reduced, and the scalability of the medical IoT system is improved.

Similarly, Guan et al. [9] combined blockchain with homomorphic encryption and searchable encryption technology to achieve secure and efficient retrieval of encrypted data while protecting the privacy of medical data. In broader IoT applications, Xu et al. [10] designed an optimization framework based on multi-authority attribute-based encryption (MA-ABE), which achieved a balance between data privacy, decentralization, scalability and storage consumption by dynamically adjusting encryption strategies. Ref. [11] proposed a blockchain data access control scheme based on the CP-ABE algorithm and applied it to the alliance blockchain Fabric. Although the scheme can effectively control user access rights and ensure the secure distribution of user attribute keys, it still has the risk of plaintext data synchronization within the channel, which may lead to data leakage. In addition, the storage capacity of the Fabric blockchain is limited, and the scheme does not support scenarios with large data volumes. Ref. [12] proposed a data access control scheme combining IPFS and blockchain technology, aiming to solve the problems of tampering, forgery and leakage that may occur in the electronic storage of law enforcement data. However, the scheme does not support fine-grained access control and cannot set fine-grained access rights for data. Ref. [13] proposed a cross-border trade data sharing and access control scheme based on blockchain, aiming to solve the problems of data security, information islands and information asymmetry faced in cross-border trade. However, the scheme does not support fine-grained access control and cannot set fine-grained access rights for data. In response to the problem of excessive access by data visitors, the scheme [14] ensures that data access behavior is traceable and verifiable by storing EMR data, visitor history records and trust values and related data on the blockchain. However, the scheme does not support data computing tasks and cannot realize computing and analysis of medical data in an encrypted state. Ref. [9] addresses the security and data island issues in IoT data sharing, using blockchain to ensure multi-party trust between enterprises, and combines fine-grained access control to solve the single point failure and data tampering problems of centralized storage. However, this solution also does not support data analysis and computing tasks. Ref. [15] proposes a privacy protection model for electronic health record anomaly detection based on CKKS fully homomorphic encryption, aiming to solve the problem of leaking patient sensitive information and diagnosis results during anomaly detection. However, this solution lacks support for data sharing with fine-grained access control.

### 2.2. Data Transaction Sharing Scheme

Jiang et al. proposed a data trading protection scheme based on blockchain. By combining AES encryption, improved homomorphic encryption technology and IPFS distributed storage, decentralized data security sharing and transaction protection were achieved, but the access control problem of data on the blockchain was not solved [16]. Zhao proposed a fair data trading protocol based on blockchain to address the challenges of data availability verification, data provider privacy protection, and payment fairness faced by data trading in the big data market. The protocol ensures the availability of transaction data, the privacy of data providers, and the fairness of both parties to the transaction, and verifies the effectiveness of the protocol through Solidity smart contracts. However, no solution was given for price negotiation of data transactions [17]. Alrawahi et al. [18], Lin et al. [19], and Cattelan et al. [20] conducted platform-based research on data transactions and traded data in the form of e-commerce transaction agreements. This method lacks flexibility and strategic interaction, and is only suitable for a limited number of scenarios, and cannot adapt to the dynamic and strategic data needs in the IoT scenario.

### 2.3. Data Flow Calculation Scheme

In scenarios such as smart cities and industrial Internet, aggregation analysis and model training of massive IoT data are the key to unlocking their value. However, outsourcing raw data containing sensitive information to cloud platforms for calculations poses a huge risk of privacy leakage. The study [21,22] conducted a security analysis of common cloud security models such as traditional SaaS, PaaS, IaaS, and CaaS. Although the services provided by third-party servers can calculate data in plain text, this is unacceptable for IoT privacy protection. As a result, many users will not choose to use cloud servers for direct calculations due to security and privacy concerns [2]. Homomorphic encryption is becoming increasingly popular in cloud computing environments because it can perform arithmetic operations on ciphertext without decryption keys, and the results are consistent with the plaintext results [23]. Cloud service providers can directly perform various calculation operations on encrypted data stored by users without decryption keys, thereby winning consumer trust and ensuring data privacy [24,25].

In addition, in recent years, Federated Learning (FL), as an emerging distributed privacy computing paradigm, has also received widespread attention in the field of IoT. The core idea of FL is to allow multiple participants to collaboratively train a machine learning model without sharing local raw data, and to protect user privacy only by exchanging model parameters. For example, the CoLearn framework proposed by Feraudo et al. [26] explored how to deploy federated learning in an IoT edge network that complies with the MUD standard, solving the engineering challenges of limited device resources and secure access. The LoByITFL scheme proposed by Xia et al. [27] starts from the algorithm level and designs a federated learning algorithm that can resist Byzantine attacks and has information-theoretic security guarantees. In addition to innovations at the algorithmic level, research has also focused on how to use blockchain technology to enhance the decentralization, security, and trustworthiness of federated learning systems. For example, Xu  et al. [28] used blockchain to build a decentralized and tamper-proof federated learning system for the Industrial Internet of Things. Similarly, the DS2PM model proposed by Chen et al. [29] also combines blockchain and federated learning to build a decentralized and trusted data sharing network.

Although federated learning excels in privacy-preserving model training, its main goal is to build shared high-performance models. This is different from the focus of this paper’s solution. The core of this work is to solve the problems of “secure transactions” and “fine-grained access control” of the data itself, aiming to provide a secure framework for data to become a trusted asset and circulate. In short, federated learning focuses on “how to use data for collaborative modeling”, while this paper’s solution focuses on “how to share and trade data itself securely and controllably”. Therefore, the two can be seen as complementary technologies: this paper’s framework can provide trusted guarantees for data transactions, and the traded data can be used as input for federated learning tasks. The comparison of this paper’s solution with other literature is summarized in Table 1.

## 3. Preliminary

This chapter will briefly introduce the core cryptographic techniques and basic concepts involved in the scheme. This background knowledge is essential for understanding the design and implementation of the system framework in subsequent chapters.

### 3.1. Blockchain Technology

Blockchain is a data sharing technology based on a distributed network. In order to ensure the consistency of all node data, the blockchain adopts a distributed consensus algorithm [30,31,32]. In recent research [33,34,35], blockchain architecture has been characterized by several key properties that are essential to its role as a decentralized ledger that ensures data integrity and traceability. Blockchain technology can be divided into three types: public chain, private chain and consortium chain, depending on the network structure and permission management. In many industries, especially in finance, medical care, and supply chain management, consortium chains have become a very suitable blockchain application solution due to their more centralized governance model and trust relationship between participants [36,37]. The core idea of smart contracts is to write the terms of traditional contracts into the blockchain in the form of code. The main advantages of smart contracts are their immutability and automated execution mechanism. Once a contract is deployed on the blockchain, no party can modify the content of the contract, ensuring the fairness and reliability of the contract terms [38,39].

### 3.2. Homomorphic Encryption Algorithm

Homomorphic encryption technology is a cryptographic technology that supports calculations on ciphertext. Data encrypted by the homomorphic encryption algorithm can be subjected to homomorphic operations. The result obtained after homomorphic decryption is the same output as the unencrypted data. According to its homomorphic properties, it can be divided into the following categories shown in Table 2.

The CKKS fully homomorphic encryption scheme was proposed by Jung Hee Cheon et al. in 2017. The scheme is based on the difficult problem RLWE [43]. It greatly improves the efficiency of approximate computing, retains high bits and discards low bits in homomorphic operations, and is widely used in scenarios such as encrypted data analysis and machine learning. This section gives a detailed theoretical description of the CKKS scheme.

The cleverness of the CKKS scheme is that it encodes the numbers that need to be calculated (such as heart rate data) into a polynomial. The encrypted polynomial is added or multiplied, and the result, after decryption, corresponds exactly to the addition or multiplication result of the original number. The relinearization key is like a “noise reducer” to control the calculation error generated after multiple rounds of multiplication to ensure the accuracy of the final result.

### 3.3. Attribute-Based Encryption Algorithm

Attribute-Based Encryption (ABE) is a cryptographic technique that allows data to be encrypted so that only users with specific attributes can decrypt and access the data. It is used to solve the problem of fine-grained access control of data in cloud storage environments and to address the problem of large-scale dynamic expansion of users. ABE systems usually involve an encryptor, a key generation center (KGC), and a decryptor. The encryptor encrypts the data, the KGC generates keys based on the user’s attributes, and the decryptor uses the key corresponding to their attributes to decrypt the data. There are two main types of ABE systems: cipher policy attribute-based encryption (CP-ABE) [44] and key policy attribute-based encryption (KP-ABE) [45].

In a CP-ABE system, the user’s private key generation is bound to the user’s attribute set, and the encryption operation is based on the access policy, that is, the encryptor specifies an access policy when encrypting data and uses the policy to encrypt the data. Only when the user’s attributes meet this policy can they decrypt the data. The basic composition of CP-ABE includes four main functions: system initialization; private key generation; data encryption and data decryption [46].

The working mechanism of CP-ABE can be compared to a “safe” with multiple keyholes. The data owner (encryptor) decides which keys (attributes, such as “cardiologist”, “Hospital A”) must be combined to open the safe, which is the “access policy”. The key generation center (KGC) distributes the keys (user private keys) owned by each user according to their identity. Only when the key set owned by a user meets the unlocking requirements set on the safe can he successfully decrypt the data.

## 4. System Structure and Design

Based on the aforementioned background and research objectives, this chapter will elaborate on the proposed blockchain secure data transaction and privacy protection framework applicable to general IoT environments. To make the design more targeted and representative, this chapter will use the Internet of Medical Things (IoMT) scenario as an example to illustrate the specific design of the system model, core modules, and algorithms.

### 4.1. Model Design

This section aims to design a secure and reliable data transaction and privacy protection model for the Internet of Things (IoT) environment. The model consists of the device layer, edge layer, data service layer, blockchain layer, and data storage layer from bottom to top. Each layer has clear responsibilities and jointly ensures the secure circulation and privacy protection of massive IoT data. The diagram of our proposed privacy protection model is depicted in Figure 1.

This multi-layered architecture enables an end-to-end secure data flow. The entire process begins at the device layer, where raw IoT data is generated by the data owner. At the edge layer, this data is encrypted using attribute-based encryption (ABE) to enforce owner-defined access policies. Subsequently, a pointer (CID) and metadata for the encrypted data are recorded on the blockchain layer, while the encrypted data itself is stored offline on the data storage layer (IPFS). Data valuation and fair pricing are managed through a game-theoretic pricing model executed by smart contracts on the blockchain. When data consumers request analysis, the data service layer coordinates third-party nodes to perform statistical analysis directly on the encrypted data, without decryption, using a secure computation scheme based on CKKS fully homomorphic encryption.

The core functions of this model are implemented by two modules, and its design is deeply integrated with the IoT scenario. The first is the data transaction sharing module, which focuses on the secure transaction of IoT data and the fine-grained sharing after the transaction, corresponding to the data transaction distributed sharing scheme supporting attribute-based encryption in Section 4.2 of this chapter. Then there is the data circulation calculation module, which focuses on the secure processing and calculation of encrypted IoT data to ensure that the privacy of the data is fully protected throughout the analysis process, corresponding to the design of the data circulation privacy algorithm based on fully homomorphic encryption in Section 4.3 of this chapter.

In this framework, the data storage layer uses the InterPlanetary File System (IPFS) to store large-scale encrypted IoT data, while the blockchain layer is responsible for recording the index and metadata of the data. The two work together to ensure the security and integrity of the data. Specifically, when the encrypted data is uploaded to the IPFS network, the system generates a unique content identifier (CID) based on its content. This CID is then recorded in the transaction of the blockchain together with meta-information such as access policies. This “on-chain index storage and off-chain data storage” model has multiple advantages. First, the decentralized nature of IPFS avoids the single point failure risk of traditional centralized storage and ensures high data availability. Second, the content addressing mechanism of IPFS means that any tampering with the data will result in a change in the CID. By anchoring the CID on the tamper-proof blockchain, this solution can provide verifiable data integrity protection. Finally, through the distributed hash table, IPFS can achieve efficient data retrieval. In summary, IPFS provides a decentralized, tamper-proof and verifiable data storage base for this framework, which forms a perfect synergy with the trust mechanism of the blockchain.

### 4.2. Data Transaction Sharing Mechanism

#### 4.2.1. Mechanism Overview

Based on the privacy protection model for the Internet of Things in the previous section, this section will specifically study the technical solutions for protecting privacy information in data transactions and circulation. The scenario of this solution is set in the Internet of Medical Things (IoMT), focusing on processing highly sensitive health data from terminals such as wearable devices and smart home health monitors. Traditional medical data platforms usually directly collect and store plaintext data from sensors (such as heart rate, blood sugar, activity level, etc.) and send it directly to third parties for sharing. This processing method makes highly personalized data extremely vulnerable to unauthorized access and malicious attacks at every stage of transmission, storage and analysis, thus seriously threatening patients’ privacy and data security. To address this issue, this section proposes a data transaction sharing mechanism designed specifically for the IoMT scenario. This mechanism aims to ensure the privacy and security of massive, high-frequency health sensor data during the transaction and sharing process through innovative cryptographic technology and decentralized architecture.

#### 4.2.2. Medical Data Transaction and Secure Sharing Based on CP-ABE

This solution is based on the sensitive data transaction and sharing scenario related to medical health. In this solution, the data consumer is defined as a an organizational entity. An entity conducts transactions with the data owner. After the data transaction is completed, this system performs privacy protection on the entire process of data sharing. The data owner uploads the data to the model, and the data consumer consumes and obtains the required data in the model. In order to achieve privacy information protection and fine-grained access control, this model uses Ciphertext-Policy Attribute-Based Encryption (CP-ABE) technology based on blockchain technology to control user access to data. Blockchain technology is used to ensure that data in this model is tamper-proof and decentralized. The overall architecture of this data transaction sharing mechanism is presented in Figure 2.

(1)System initialization: The master key (MK) and public parameters (PP) are generated during the system initialization phase. Public parameters are used for data encryption and key generation, and the master key is used to generate user private keys (the detailed process is described in Algorithm A1 in Appendix A).(2)Data consumer key generation algorithm: The algorithm generates a private key for the data consumer that is bound to its attribute set.The key generated by the algorithm and bound to the attribute set can only be used to decrypt and obtain data that matches the attribute set, but cannot decrypt other data that does not match (the detailed steps are shown in Algorithm A2 in Appendix A).(3)Data encryption algorithm of the data owner: In this algorithm, the data owner generates ciphertext based on the access policy and the original data M. The access policy is in the form of an access tree, which is defined according to the sensitivity and usage requirements of the data (this encryption process is formalized in Algorithm A3 in Appendix A).(4)Data consumer decryption algorithm: In this algorithm step, the data consumer uses the private key SK to decrypt the ciphertext CT and obtain the data M (the detailed decryption algorithm can be found in Algorithm A4 in Appendix A).

#### 4.2.3. Data Trading Technology Solutions

This data trading solution first designs a comprehensive value assessment method for the data provided by the data provider, which can make a preliminary assessment of the data for subsequent data pricing. Then the game pricing method is used to confirm the final pricing.

In the medical privacy scenario, the integrity of user health data is one of the important dimensions for measuring data quality. This solution evaluates the integrity of medical health data by detecting the residual of the data, and combines the non-linear state estimation model and weighted least squares method for quantitative analysis. Assume that the original health data is data collected directly from medical equipment or users (such as blood pressure, heart rate, blood sugar, etc.), the state vector is a variable describing the user’s health status, and the measurement error covariance matrix describes the statistical characteristics of the measurement error. The following is a detailed measurement method and formula explanation:(1)In the context of medical health data, construct a non-linear state estimation model to describe the relationship between health data and the system state, as formalized in Equation (Equation 1):(1)z=n(s)+eThis model is used to predict health data and compare it with actual collected data, thereby assessing data integrity. Here, $s\in R^{M}$ is the state vector, which includes the user’s health status (e.g., blood sugar levels, body temperature). $z\in R^{N}$ is the measurement vector, representing the actually collected health data. n(s) is a non-linear function related to the measurement and state vectors, describing the relationship between them. *e* is the error vector, representing measurement error or noise, typically assumed to be e∼N(0,C), where $C\in R^{N\times M}$ is the covariance matrix of the measurement error. This formula means that the actual measured health data *z* should be approximately equal to a theoretical prediction value n(s) based on the user’s actual health condition *s*, plus some random measurement errors *e*.(2)To estimate the system state vector *s*, use the Weighted Least Squares (WLS) method to minimize the residual between the measured values and the model’s predicted values. The objective function is defined in Equation (Equation 2):(2)$$\hat{s}=\arg\min_{s}{\left[z-n(s)\right]}^{T}C^{-1}\left[z-n(s)\right]$$The goal of this formula is to find the best estimate $\hat{s}$ that minimizes the gap between the theoretical prediction and the actual measurement by adjusting the estimate of the user’s true health status *s*.This is solved using the Gauss–Newton iteration method, where the state vector is updated in each iteration according to Equation (Equation 3):(3)$$\Delta s={\left[J{\left(s\right)}^{T}C^{-1}J\left(s\right)\right]}^{-1}J{\left(s\right)}^{T}C^{-1}\left(z-n\left(s\right)\right)$$Here, J(s) is the Jacobian matrix, representing the partial derivative of the non-linear function n(s) with respect to the state vector *s*, and $\hat{s}$ is the optimal state estimate.(3)The core of integrity detection is to assess data integrity by calculating the residual. The residual is defined as the difference between the actual measured value and the model’s predicted value, as defined in Equation (Equation 4):(4)$$∥z-n\left(\hat{s}\right){∥}_{2}$$The “residual” here is the difference between the actual measured data and our best prediction. If this difference is too large, it means that the original data may be missing or tampered with, and the integrity is low.If the residual exceeds a predefined detection threshold ψ, the data is considered incomplete. The detection threshold ψ can be set based on the statistical properties of the medical data or clinical standards.(4)The integrity value, denoted as Int(s), quantifies the data integrity using the residual and the detection threshold, as shown in Equation (Equation 5):(5)$$Int\left(s\right)=\lambda (∥z-n\left(\hat{s}\right){∥}_{2}-\psi )$$Here, λ is an adjustment parameter used to map the residual to the range of the integrity value. If $∥z-n\left(\hat{s}\right){∥}_{2}\leq \psi$, the data is considered complete, and the integrity value is high. If $∥z-n\left(\hat{s}\right){∥}_{2}>\psi$, the data is considered incomplete, and the integrity value is low.(5)To combine the integrity value with the results of other valuation dimensions (such as timeliness valuation), the integrity value needs to be normalized. Here, $Int_{\min}$ and $Int_{\max}$ are the minimum and maximum possible values of the integrity value, respectively. These are typically set based on historical data or domain knowledge. The normalization formula is given by Equation (Equation 6):(6)$$Int_{nor}=\frac{Int\left(s\right)-Int_{\min}}{Int_{\max}-Int_{\min}}$$

The data volume parameter, *D*, is used to measure the size of the data. Typically, the larger the data volume, the higher its value. The data volume parameter can be defined as $D=Q/Q_{unit}$, where *Q* is the actual data volume (measured by the scale of the dataset, such as the number of individuals and attributes included) and $Q_{unit}$ is the size of a unit dataset, which can be set based on domain knowledge.

The authority parameter, *A*, is used to measure the authority and credibility of the data provider. It is set by the pricing module for the data provider and can be dynamically adjusted based on the provider’s historical data. A higher authority level generally corresponds to a higher data value. The authority parameter *A* has a value range of [0,1], which facilitates its combination with other parameters. By combining the integrity, data volume, and provider authority, the value of medical health data can be comprehensively assessed. The comprehensive valuation function *V* is defined in Equation (Equation 7):(7)$$V=\Gamma \cdot Int_{nor}\cdot D\cdot A$$

It is important to acknowledge that the weighting factors and parameters (e.g., Γ, λ, $Int_{\min}$, $Int_{\max}$) in this valuation function are presented here in a generalized form. In this research’s prototype implementation and theoretical analysis, these parameters are treated as configurable variables based on domain knowledge, a necessary simplification for establishing the underlying model. However, in a real-world deployment, these factors must be empirically derived and calibrated. A production-grade system would require an upfront data analysis phase to determine the optimal values for these parameters using historical market data from a specific domain (e.g., cardiology data transactions). These weighting factors can be derived by using techniques such as regression analysis or machine learning to model the relationship between data features and their final transaction price. Therefore, calibrating this pricing model using domain-specific data is a key step in its practical application and an important future direction.

#### 4.2.4. Dynamic Game-Theoretic Pricing

After assessing the intrinsic data value *V* using Equation (Equation 7), the system employs a multi-objective optimization game-theoretic pricing model to negotiate and determine the final transaction price Φ. This model aims to balance the dual objectives of the data provider, who seeks to maximize profit, and the data consumer, who wishes to acquire high-quality data at a reasonable cost.

In this game model, the utility functions for both the data provider and the data consumer are explicitly defined. For the data provider, the total utility $U_{Provider}$ consists of three components: revenue from data sales, minus the data collection costs $C_{tra}$, and minus the data processing costs $C_{pro}$. The sales revenue is determined by the final price Φ, while the processing cost is related to the data size *d*. The utility function is formalized as Equation (Equation 8):(8)$$U_{Provider}=\sum_{t\in T_{a}}\phi_{j,t}-C_{tra}\left(V_{j}\right)-C_{pro}\left(\Sigma d_{j,t}\right)$$
For the data consumer, the utility $U_{Consumer}$ depends on the comprehensive value *V* of the acquired data and the purchasing cost Φ. By introducing adjustment parameters $p_{1}$ and $p_{2}$ to balance different dimensions of utility, the utility function is defined in Equation (Equation 9):(9)$$U_{Consumer}=p_{1}\sum_{t\in T_{a}}\phi_{i,t}-p_{2}V_{j}$$
The core of this pricing game is a multi-objective optimization problem that aims to maximize the utility of the provider while minimizing the cost of the consumer. It is specifically Equation (Equation 10):(10)$$\left\{\begin{array}{c} \max U_{Provider}\\ \min U_{Consumer} \end{array}\right.$$
To ensure fair and feasible transactions, the algorithm establishes a reasonable price range $\left[\Phi_{lower},\Phi_{upper}\right]$ and a data size range $\left[d_{req},d_{max}\right]$. All bids must adhere to these constraints in Equation (Equation 11):(11)$$\left\{\begin{array}{c} \phi_{lower}\leq \phi_{j,t}\leq \phi_{upper}\\ d_{req}\leq d_{j,t}\leq d_{max} \end{array}\right.$$
Finally, a multi-objective optimization algorithm (e.g., Pareto optimality search) is employed to find the set of equilibrium solutions that satisfy the constraints, and the final equilibrium price $\Phi_{final}$ is determined through negotiation. This game-theoretic pricing scheme facilitates fair data transactions at a mutually agreeable price by maximizing the utility for both parties.

### 4.3. Data Circulation Privacy Algorithm Based on Fully Homomorphic Encryption

#### 4.3.1. Program Overview

Based on the privacy protection model for the Internet of Things in Section 1, this section elaborates on the data circulation privacy algorithm based on fully homomorphic encryption (FHE). This solution proposes a data processing solution based on homomorphic encryption technology to meet the privacy computing needs of IoT sensor data. This solution takes medical and health as the research scenario and uses continuous physiological indicators (such as floating-point sequences of heart rate, body temperature, and blood pressure) from wearable devices or smart home sensors as source data. These encrypted sensor data are securely stored in distributed storage nodes (IPFS). When data analysis is required, the computing requester (such as researchers or AI models) publishes the computing task (such as “calculating the average heart rate for the past 24 h” or “analyzing the volatility of body temperature data”) to a third-party computing node through the blockchain. The third-party computing node directly performs homomorphic operations on the encrypted floating-point sensor data, making it possible to perform complex statistical analysis without decryption.

It is worth noting that the adoption of the CKKS-based fully homomorphic encryption scheme does bring significant computational overhead. Other cutting-edge paradigms, such as federated learning (FL) and encoded computing, can indeed provide strong privacy guarantees at a lower computational cost in specific scenarios, even reaching information-theoretic security levels. However, the choice of homomorphic encryption is based on the specific core requirements of the data circulation module in this framework. First, in terms of functional goals, the scheme aims to support third parties to directly perform general statistical analysis (e.g., calculating mean, variance) on encrypted raw sensor data, which is very different from the goal of federated learning that focuses on collaborative training models. Second, in terms of computational flexibility, the CKKS scheme uniquely supports arithmetic operations on encrypted floating-point vectors, thereby enabling complex computational tasks to be performed on sensitive data without decryption. This degree of flexibility is generally not available in many linear-computation-specific encoding schemes or federated learning frameworks that only aggregate model parameters.Therefore, if homomorphic encryption is replaced with other technologies, the core capability of the scheme—that is, encrypted data can be securely processed and analyzed by a trusted third party while maintaining end-to-end confidentiality will be fundamentally lost. It can be argued that accepting higher computational costs is a necessary trade-off in order to gain this high computational flexibility and strong privacy protection for direct data analysis.

#### 4.3.2. Key Algorithms

The CKKS fully homomorphic encryption algorithm is the core technology for data circulation computing, supporting addition and multiplication operations on encrypted data without decrypting the data. The specific algorithm is as follows:(1)Key Generation Algorithm $\mathrm{KeyGen}\left(N,q,\chi \right)\to \left(b,a,s,b^{\prime },a^{\prime }\right)$In the initialization phase of the CKKS algorithm, the Public Key (PK), Secret Key (SK), and Relinearization Key (RK) are generated. The process begins with parameter selection, choosing a polynomial ring $R=Z\left[X\right]/\langle X^{N}+1\rangle$ with degree *N*, a modulus *q*, and a noise distribution χ (typically Gaussian). These parameters collectively determine the algorithm’s security and computational capability. A secret key *s* is then randomly sampled from the distribution χ. This secret key is a core component used for decryption and public key generation. To create the public key, this paper samples a random polynomial $a\in R_{q}$ and a noise term e←χ, then compute the first public key component *b* using Equation (Equation 12):(12)b=−a·s+e(modq)The resulting public key (b,a) ensures data confidentiality during transmission and computation. To generate the relinearization key, this paper samples another random polynomial $a^{\prime }\in R_{q^{2}}$ and a noise term $e^{\prime }\leftarrow \chi$. Using a scaling factor *p*, the first relinearization key component $b^{\prime }$ is then computed as shown in Equation (Equation 13):(13)$$b^{\prime }=-a^{\prime }\cdot s+e^{\prime }+p\cdot s^{2}\;\left(\mathrm{mod}\;q^{2}\right)$$The relinearization key $\left(b^{\prime },a^{\prime }\right)$ is essential for supporting homomorphic multiplication operations, thereby ensuring the correctness and security of the computational results.(2)Data Encryption Algorithm $\mathrm{Encrypt}\left(m,\left(b,a\right)\right)\to \left(c_{0},c_{1}\right)$:The data encryption algorithm first encodes the plaintext data into a polynomial vector and then encrypts it using the public key to generate a ciphertext. The first step is encoding, where a floating-point vector $m=(m_{1},m_{2},\ldots ,m_{k})$ is encoded into a polynomial $m\left(x\right)\in Z_{q}\left[x\right]/\langle x^{N}+1\rangle$. This process maps the floating-point numbers onto the polynomial ring for encryption and computation. The second step is plaintext encryption. Sampling a random polynomial v←χ and noise terms $e_{0},e_{1}\leftarrow \chi$. The ciphertext $\left(c_{0},c_{1}\right)$ is then computed according to the formulas in Equation (Equation 14):(14)$$\begin{array}{cc} ccc_{0} & =v\cdot b+e_{0}+m\left(x\right)\;\left(\mathrm{mod}\;q\right)\\ c_{1} & =v\cdot a+e_{1}\;\left(\mathrm{mod}\;q\right) \end{array}$$The resulting ciphertext $\left(c_{0},c_{1}\right)$ is the encrypted data, which allows for homomorphic computations to be performed without decryption.(3)Homomorphic Addition $Add\left(ct,ct^{\prime }\right)\to ct_{add}$:Homomorphic addition performs an addition operation on two ciphertexts, $ct=(c_{0},c_{1})$ and $ct^{\prime }=\left(c_{0}^{\prime },c_{1}^{\prime }\right)$, to generate a new ciphertext, $ct_{add}$. The decrypted value of the resulting ciphertext is equal to the sum of the plaintexts corresponding to the original ciphertexts. Given two ciphertexts, which correspond to two encrypted floating-point vectors, the addition is computed directly on the ciphertext components, as shown in Equation (Equation 15):(15)$$\begin{array}{cc} ccc_{add1} & =c_{0}+c_{0}^{\prime }\;\left(\mathrm{mod}\;q\right)\\ c_{add2} & =c_{1}+c_{1}^{\prime }\;\left(\mathrm{mod}\;q\right) \end{array}$$The result is the ciphertext $ct_{add}=\left(c_{add1},c_{add2}\right)$. The homomorphic addition operation is performed directly on the ciphertexts without needing to decrypt the data, thus ensuring privacy during the computation process.(4)Homomorphic Multiplication $Multiply\left(ct,ct^{\prime },\left(b^{\prime },a^{\prime }\right)\right)\to ct_{mul}$:Homomorphic multiplication performs a multiplication operation on two ciphertexts to generate a new ciphertext. Since the multiplication operation introduces additional noise, it is necessary to use the relinearization key for noise management after the multiplication is complete. The process begins with two input ciphertexts, $ct=(c_{0},c_{1})$ and $ct^{\prime }=\left(c_{0}^{\prime },c_{1}^{\prime }\right)$, which correspond to two encrypted floating-point vectors. The multiplication operation first computes an intermediate ciphertext $\left(c_{a},c_{b},c_{c}\right)$ using the formulas in Equation (Equation 16):(16)$$\begin{array}{cc} ccc_{a} & =c_{0}\cdot c_{0}^{\prime }\;\left(\mathrm{mod}\;q\right)\\ c_{b} & =c_{0}\cdot c_{1}^{\prime }+c_{1}\cdot c_{0}^{\prime }\;\left(\mathrm{mod}\;q\right)\\ c_{c} & =c_{1}\cdot c_{1}^{\prime }\;\left(\mathrm{mod}\;q\right) \end{array}$$Next, relinearization is performed using the relinearization key $\left(b^{\prime },a^{\prime }\right)$ to transform the above ciphertext into a linear form as shown in Equation (Equation 17):(17)$$\begin{array}{cc} ccc_{mul0} & =c_{a}+b^{\prime }\cdot c_{c}\;\left(\mathrm{mod}\;q\right)\\ c_{mul1} & =c_{b}+a^{\prime }\cdot c_{c}\;\left(\mathrm{mod}\;q\right) \end{array}$$The homomorphic multiplication operation is performed on the ciphertexts, yielding a new ciphertext result $ct_{mul}=\left(c_{mul0},c_{mul1}\right)$. The relinearization step ensures the correctness and security of the multiplication result.(5)Data Decryption Algorithm $Decrypt(\left(c_{0},c_{1}\right),s)\to m$:Data decryption uses the secret key to decrypt the ciphertext and recover the plaintext data. The algorithm takes a ciphertext $\left(c_{0},c_{1}\right)$ and a secret key *s* as input. The decryption operation computes the resulting polynomial m(X), which restores the ciphertext to its polynomial form using Equation (Equation 18):(18)$$m\left(X\right)=c_{0}+c_{1}\cdot s\;\left(\mathrm{mod}\;q\right)$$Finally, plaintext decoding is performed, where the polynomial m(X) is decoded into a floating-point vector $m=(m_{1},m_{2},\ldots ,m_{k})$. This decoding process maps the polynomial back to the original floating-point vector.(6)Rescale Algorithm $Rescale\left(\left(c_{0},c_{1}\right),p\right)\to \left(c_{0}^{\prime },c_{1}^{\prime }\right)$:The CKKS algorithm uses a rescaling technique to ensure the precision and reliability of computation results. Each homomorphic operation (such as addition and multiplication) introduces additional noise, so periodic rescaling operations are necessary to reduce the noise level and maintain computational precision. To rescale a ciphertext $\left(c_{0},c_{1}\right)$, this paper computes $c_{0}^{\prime }=\left\lfloor c_{0}/p\right\rfloor$ and $c_{1}^{\prime }=\left\lfloor c_{1}/p\right\rfloor$. Here, *p* is the scaling factor, and ⌊·⌉ denotes the rounding operation. The pair $\left(c_{0}^{\prime },c_{1}^{\prime }\right)$ is the rescaled ciphertext. The rescaling operation lowers the noise level, thereby ensuring the precision of the computation results.

#### 4.3.3. Technical Solution

This section describes in detail the phased process of the data circulation privacy algorithm based on CKKS fully homomorphic encryption, in order of priority: initialization and key generation of the key management center, data encryption upload, ciphertext calculation of the third-party computing center, decryption operation, and realizes the secure circulation and calculation of data in an encrypted state. The application scenario of this solution is defined in the medical and health scenario. The entities involved in the solution include the computing requester DR, the key management center KM, the third-party computing center, the blockchain, and IPFS. The following is the specific process:(1)Initialization PhaseIn this phase, the Key Management center performs the initialization and key generation for the CKKS fully homomorphic encryption scheme. When a computation requester submits a computation task through a secure channel, the Key Management center first creates a globally unique task identifier (TaskID) for this task. This TaskID will serve as the index key for the entire computation lifecycle, used for subsequent key management, task tracking, and result association. The system returns this TaskID to the requester as a task acceptance certificate.When the computation requester submits the task, they are assigned the corresponding TaskID to index it. The parameters for the CKKS scheme are selected, including the dimension of the polynomial ring *N*, the ciphertext modulus *q*, and the scaling factor Δ. The context for this CKKS scheme is generated, denoted as ctx.Key generation then proceeds. The private key sk and public key pk for the computation requester are generated, along with the relinearization key rlk for ciphertext relinearization. This is represented as (pk,sk)←CKKS.KeyGen(ctx) and rlk←CKKS.RelinKeyGen(ctx,sk).To enhance security and support efficient computation, the system additionally generates a symmetric key $k_{T}$ for encrypting and decrypting the computation task. This key will be used to protect the specific content of the computation task, providing an extra layer of encryption during transmission and storage.Finally, the keys are distributed. The public key pk, relinearization key rlk, and the symmetric key $k_{T}$ are distributed to the computation requester and the third-party computation node. The private key sk is distributed only to the computation requester.(2)Task Upload PhaseIn this phase, the computation requester preprocesses the data and uploads the encrypted computation task.The computation owner preprocesses the data *M*, which is a matrix representing a dataset of health indicators for multiple individuals. Assuming there are *n* individuals, each with *d* attributes, the matrix *M* can be represented as follows, where $m_{i,j}$ denotes the *j*-th attribute value for the *i*-th individual, as represented in Equation (Equation 19):(19)$$M=\left[\begin{array}{cccc} m_{1,1} & m_{1,2} & \ldots & m_{1,d}\\ m_{2,1} & m_{2,2} & \ldots & m_{2,d}\\ \vdots & \vdots & \ddots & \vdots \\ m_{n,1} & m_{n,2} & \ldots & m_{n,d} \end{array}\right]$$Next, the data is encrypted. Each element $m_{i,j}$ of the matrix *M* is homomorphically encrypted, generating the corresponding ciphertext $ct_{i,j}$. This is done using the CKKS scheme’s encryption algorithm, formally expressed as $ct_{i,j}\leftarrow CKKS.Encrypt\left(pk,m_{i,j}\right)$.After encryption, the original plaintext matrix *M* is transformed into a ciphertext matrix CT, represented as follows, where $ct_{i,j}$ is the ciphertext of $m_{i,j}$, as shown in Equation (Equation 20):(20)$$CT=\left[\begin{array}{cccc} ct_{1,1} & ct_{1,2} & \ldots & ct_{1,d}\\ ct_{2,1} & ct_{2,2} & \ldots & ct_{2,d}\\ \vdots & \vdots & \ddots & \vdots \\ ct_{n,1} & ct_{n,2} & \ldots & ct_{n,d} \end{array}\right]$$The encrypted data is then uploaded to IPFS. The encrypted ciphertext matrix CT is serialized into a file and uploaded. After the upload, IPFS generates a unique Content Identifier (CID) for each file.The CID, metadata (such as requester’s identity, request time, data dimensions, etc.), and the computation task (e.g., the computation function *f* and the involved attributes) are bundled together, denoted as TP (Task Package). The national standard SM4 algorithm is used to encrypt the entire task package, resulting in the ciphertext form $CT_{TP}$. This is represented as $CT_{TP}=SM4.encrypt\left(k_{T},TP\right)$.Finally, the computation request is uploaded to the blockchain. The computation request, including metadata and the computation function, is submitted to the Hyperledger Fabric blockchain along with the TaskID and $CT_{TP}$. The transaction ID for this submission is then obtained.(3)Ciphertext Computation PhaseIn this phase, the third-party computation center retrieves the computation task and performs homomorphic computation on the ciphertext.The third-party computation center retrieves the encrypted computation task description, $CT_{TP}$, from the blockchain using the Task_ID. The $CT_{TP}$ is decrypted to obtain the task package TP, from which the specific computation tasks are parsed, including the data CID, metadata, and the description of the computation function *f*. This is represented as $TP=SM4.decrypt(k_{T},CT_{TP})$.Using the CID, the ciphertext CT is retrieved from IPFS, represented as CT=IPFS.download(CID).Based on the description of the computation function *f*, the required operations are parsed. These could be computation tasks such as calculating the mean, variance, or correlation analysis as mentioned above. Homomorphic operations are performed on the elements of the ciphertext matrix according to the defined algorithms. The next subsection will provide a detailed explanation of the three computation tasks mentioned, resulting in the final ciphertext $CT_{result}$.The computation result, $CT_{result}$, is uploaded to IPFS, which returns a content identifier, $CID_{result}$. This is represented as $CID_{result}\leftarrow IPFS.upload\left(CT\right)$.Finally, the resultID and $CID_{result}$ are uploaded to the Fabric blockchain, and the transaction ID (txId) for this on-chain transaction is obtained.(4)Result Retrieval PhaseIn this phase, the computation requester retrieves and decrypts the computation result.The computation requester retrieves the computation result ciphertext CID, $CID_{result}$, from the blockchain using the resultID. Based on $CID_{result}$, the homomorphic ciphertext of the computation result, $CT_{result}$, is retrieved from IPFS. This is represented as $CT_{result}=IPFS.download\left(CID_{result}\right)$.Using the private key sk, the requester decrypts $CT_{result}$ to obtain the plaintext result. This is represented as $result\leftarrow CKKS.Decrypt(CT_{result},sk)$.The computation requester successfully obtains the computation result, result.

Through the above steps, the mean, variance and attribute correlation analysis can be calculated for the encrypted data. And this solution makes full use of the homomorphic characteristics of CKKS, allowing third-party data centers to handle a wider range of computing requests, including some more complex calculations, while protecting data privacy and supporting complex statistical analysis tasks, which is very suitable for sensitive scenarios such as medical data.

## 5. Security Analysis

This chapter aims to conduct a comprehensive and in-depth security analysis of the proposed scheme. Before the analysis, the key security assumptions in the scheme are first explained, especially the source of randomness. The security of this scheme depends largely on the unpredictability of random numbers in multiple cryptographic algorithms. For example, α, β used to generate the master key in Algorithm A1, and r, $r_{j}$ used to generate the user’s private key in Algorithm A2, must be cryptographically secure random numbers. To this end, this scheme makes the following assumptions: All random numbers are generated by relying on a cryptographically secure pseudo-random number generator (CSPRNG). The generator must meet the “next bit unpredictability” principle, that is, its output is computationally indistinguishable from a true random sequence.

### 5.1. Threat Model and Security Goals

A formal security model consists of three parts: system entities, threat model, and security goals. First, the ecosystem of this solution mainly includes the following four entities: data owners, data consumers, blockchain networks, and IPFS networks. Second, this paper considers a strong threat model. Assume that there is an adversary A whose capabilities include: (1) Network attack capabilities: Adversary A can eavesdrop on, intercept, and replay communications between all entities in the network; (2) Internal entities: Data consumers, blockchain nodes, and IPFS nodes are all assumed to be internal entities. They will comply with the protocol process, but will try to infer additional privacy from all the information they obtain; (3) User collusion: Malicious data consumers who do not comply with the access policy may collude and try to combine their private keys to decrypt the data. Based on the above threat model, this solution aims to achieve the following core security goals:Data Confidentiality: Encrypted data stored on IPFS or transmitted in the network should be computationally indistinguishable from random numbers for any entity that does not hold a set of attributes that satisfy the access policy.Access Control: Only authorized users with a set of attributes that satisfy the access policy defined by the data owner can successfully decrypt and access the data.

### 5.2. Security Proofs

This section demonstrates how the proposed scheme satisfies the security goals defined above.

#### 5.2.1. Data Confidentiality

The data confidentiality of the scheme is ensured by the underlying cryptographic primitives, namely the CP-ABE scheme for data sharing and the CKKS scheme for data circulation computation.

For the data sharing phase, the confidentiality relies on the standard security notion of Ciphertext-Policy Attribute-Based Encryption (CP-ABE), which is formally defined under the model of Indistinguishability under Selective-Attribute Set and Chosen-Plaintext Attack (IND-sAtt-CPA). This model is formalized through a security game between a challenger C and a polynomial-time adversary A. The  game proceeds as follows:Init: The adversary A chooses a challenge access structure $A^{*}$ and sends it to C.Setup: The challenger C runs the Setup algorithm to generate public parameters PK and a master key MK, sending PK to A.Query Phase 1: A makes a series of private key queries for attribute sets *S*, with the restriction that no queried set *S* can satisfy the challenge policy (i.e., $A^{*}\left(S\right)\neq 1$). For each valid query, C provides the corresponding private key $SK_{S}$.Challenge: A submits two equal-length messages, $m_{0}$ and $m_{1}$. C randomly chooses a bit b∈{0,1}, encrypts $m_{b}$ under $A^{*}$ to create the challenge ciphertext $C^{*}=\mathrm{Encrypt}\left(PK,m_{b},A^{*}\right)$, and sends $C^{*}$ to A.Query Phase 2: A continues making private key queries under the same restriction.Guess: Finally, A outputs a guess bit $b^{\prime }$.

A CP-ABE scheme is considered IND-sAtt-CPA secure if, for any polynomial-time adversary A, its advantage in winning this game is negligible. The advantage is defined in Equation (Equation 21):(21)$$Adv_{CP-ABE,A}^{\mathrm{IND}-\mathrm{sAtt}-\mathrm{CPA}}\left(k\right)=\left|\mathrm{Pr}\left[b^{\prime }=b\right]-\frac{1}{2}\right|\leq negl\left(k\right)$$
where *k* is the security parameter. This definition intuitively states that the adversary’s ability to guess which message was encrypted is not significantly better than a random guess.

The security of the CP-ABE construction employed in this framework relies on the Decisional Bilinear Diffie–Hellman (DBDH) Assumption. This assumption is defined over two cyclic groups $G_{1}$ and $G_{2}$ of prime order *p* with a generator $g\in G_{1}$ and a bilinear map $e:G_{1}\times G_{1}\to G_{2}$. The DBDH assumption asserts that no polynomial-time algorithm can distinguish between the random tuple distribution $\left(g,g^{a},g^{b},g^{c},e{\left(g,g\right)}^{abc}\right)$ and the random element distribution $\left(g,g^{a},g^{b},g^{c},Z\right)$, where $a,b,c\in Z_{p}$ are random and $Z\in G_{2}$ is random. In other words, given $\left(g,A=g^{a},B=g^{b},C=g^{c}\right)$, it is computationally hard to decide if a given $T\in G_{2}$ equals $e{\left(A,B\right)}^{c}$.

The IND-sAtt-CPA security of the proposed CP-ABE scheme can now be argued via a reduction to the DBDH problem. It can be proven that if there exists a polynomial-time adversary A that can break the IND-sAtt-CPA security of the scheme with a non-negligible advantage ϵ, then an algorithm B can be constructed that uses A as a subroutine to solve the DBDH problem with a non-negligible advantage. The proof proceeds as follows: B receives a DBDH challenge instance $\left(g,A=g^{a},B=g^{b},C=g^{c},T\right)$ and must decide if $T=e{\left(g,g\right)}^{abc}$. B simulates the IND-sAtt-CPA game for A. During the Setup phase, B embeds the DBDH challenge elements into the public parameters PK, for instance, by setting parts of the master key to involve $g^{a}$. During the Key Query phase, for any attribute set *S* not satisfying the challenge policy $A^{*}$, B can generate a valid private key $SK_{S}$ without knowing the full master key. This is possible due to the properties of linear secret sharing schemes. For the Challenge phase, B constructs the challenge ciphertext $C^{*}$ by setting a key component to $C^{\prime }=m_{b}\cdot T$. If $T=e{\left(g,g\right)}^{abc}$, then $C^{*}$ is a valid ciphertext of $m_{b}$. If *T* is random, the information about $m_{b}$ is perfectly hidden from A. If A guesses $b^{\prime }$ correctly ($b^{\prime }=b$), B guesses that $T=e{\left(g,g\right)}^{abc}$; otherwise, it guesses *T* is random. The advantage of B in solving the DBDH problem can be shown to be ϵ/2. Since ϵ is non-negligible, so is ϵ/2. This contradicts the DBDH assumption, thus proving that no such adversary A can exist, and the CP-ABE scheme is IND-sAtt-CPA secure.

#### 5.2.2. Access Control

The scheme’s fine-grained access control is inherently provided by the cryptographic mechanisms of CP-ABE. The access control model involves several key entities: the Data Owner, the Data User, and the Attribute Authority (AA). The AA is an authoritative entity responsible for verifying user identities and issuing attributes and corresponding private keys. The entire system operates based on a bilinear pairing setting $\left(p,G_{0},G_{T},e,g\right)$, where $G_{0},G_{T}$ are groups of prime order *p* and $e:G_{0}\times G_{0}\to G_{T}$ is the pairing map.

The workflow is formalized through four core algorithms, as detailed in Appendix A:Setup: The AA runs $Setup(1^{k})$ to generate the public key PK and the master key MK. A typical construction would yield $PK=(g,h=g^{\beta },e{\left(g,g\right)}^{\alpha })$ and $MK=(\beta ,g^{\alpha })$, where $\alpha ,\beta \in Z_{p}$ are random exponents.Key Generation: For a user with an attribute set *S*, the AA performs KeyGen(MK,S) to compute the private key $SK_{S}$. This involves choosing random values to generate a key tuple that binds the user’s identity to their attributes.Encryption: When a data owner shares a message *M*, they define an access policy A and compute the ciphertext CT←Encrypt(PK,M,A). The ciphertext structure embeds this access policy.Decryption: A user can successfully decrypt the ciphertext to recover the message *M* if and only if their attribute set *S* satisfies the policy A, i.e., A(S)=1.

The policy expressiveness of this mechanism stems from the underlying Linear Secret Sharing Scheme (LSSS). An access policy A, often represented as an access tree, is converted into a share-generating matrix (A,ρ), where *A* is an l×n matrix and ρ maps each row $A_{i}$ to an attribute ρ(i). To encrypt under this policy, the encryptor chooses a secret sharing vector $v={\left(s,v_{2},\ldots ,v_{n}\right)}^{T}\in Z_{p}^{n}$ and calculates the shares $\lambda_{i}=A_{i}\cdot v$ for i=1,…,l. The ciphertext components are then computed as $C_{i}=g^{\lambda_{i}}$ and $C_{i}^{\prime }=H{\left(\rho \left(i\right)\right)}^{\lambda_{i}}$.

A user with an attribute set *S* can decrypt if they can find a set of constants ${\left\{\omega_{i}\in Z_{p}\right\}}_{i\in I}$ (where I={i|ρ(i)∈S}) such that $\sum_{i\in I}\omega_{i}A_{i}=\left(1,0,\ldots ,0\right)$. This allows them to combine key and ciphertext components through pairings to cancel out the randomness and reconstruct the secret. By combining the intermediate values with the constants $\omega_{i}$, the term $e{\left(g,g\right)}^{rs}$ can be reconstructed, and finally, the recovery message is calculated using Equation (Equation 22):(22)$$\frac{\tilde{C}}{\prod_{i\in I}{\left(e\left(K_{\rho (i)},C_{i}\right)/e\left(K_{\rho (i)}^{\prime },C_{i}^{\prime }\right)\right)}^{\omega_{i}}}=M$$

This expression simplifies to *M* only if the denominator correctly reconstructs $e{\left(g,g\right)}^{\alpha s}$, which requires the user’s attributes to satisfy the policy. Consequently, any user or group of colluding users whose combined attributes do not satisfy the access policy cannot reconstruct the secret, thus enforcing fine-grained access control.

## 6. Experiments

The test environment configuration of the system proposed in this paper is as follows. The hardware consists of an Intel(R) Xeon(R) Platinum 8369B CPU, 16 GB of memory, and a 1 TB SSD. The software environment includes CentOS Linux release 7.9.2009 (Core), Java 1.8, MySQL 5.7.44, Microsoft SEAL 3.7.3, Hyperledger Fabric v2.2.1, and IPFS Kubo 0.33.2. The backend of the prototype system of this solution is built on Hyperledger Fabric v2.2.1 and deployed in containers through Docker and Docker Compose. The network topology consists of two independent peer organizations (org1 and org2), each of which has two peer nodes responsible for transaction endorsement and ledger maintenance. The sorting service uses the Raft co-SID protocol recommended in the production environment to ensure the consistency and fault tolerance of transaction sorting. To ensure the independence of identity management, each organization is configured with a dedicated certificate authority (CA) server. In addition, all peer nodes are configured to use CouchDB as the state database to support rich queries on ledger data. All smart contracts (chain codes) are written in Go language, and the channel configuration uses the default parameters of Fabric.

### 6.1. Functional Testing

In order to verify the functional integrity and correctness of the prototype system, a series of end-to-end unit and integration functional tests are performed in this section. The test method is designed to strictly simulate a complete data sharing and computing process in the application scenario of the Internet of Medical Things (IoMT), from the authentication and joining of new institutions (users) to the final secure access to encrypted medical data. The purpose of the test is to ensure that each core module in the system works as expected and can work together to complete the task safely and reliably. The specific test items and their successful verification results are shown in Table 3. The following will elaborate on the methods used for each test item and its specific relevance in the medical scenario:Fabric node registration: This test simulates a new medical institution or research center joining the network. By calling the fabric-ca-client tool, it first creates an administrator identity for the new organization and issues it an X.509 certificate and private key. Subsequently, this administrator identity is used to register and register the identity of a new user (for example, a doctor), and verify whether the user can successfully obtain his certificate and private key pair. In the medical scenario, this is the access basis for the entire trusted medical data ecosystem. It ensures that only authenticated and legitimate medical institutions and personnel (such as hospitals, doctors, and researchers) can enter the network, providing identity security for subsequent data transactions and sharing.IPFS data upload/download and Fabric chain query: This test simulates a wearable device (data owner) uploading its encrypted health data file. During the test, the file is first uploaded to the IPFS node and a unique content identifier (CID) is returned to verify whether the operation is successful. Subsequently, the CID and related metadata are submitted to the Fabric ledger by calling the putValue chain code function. Finally, the CID is queried and obtained by calling the getValue chain code function, and then the original encrypted file is downloaded from the IPFS network using this CID to verify the integrity of the process.Data transaction and encryption sharing process: The test starts with the data owner uploading an encrypted health report metadata and setting the expected price. Subsequently, a data consumer is simulated to bid for the data, and the system is verified to be able to calculate the optimal price based on data valuation and game pricing algorithm for confirmation by both parties. After both parties confirm the transaction, the test process verifies whether the data owner can successfully set the access policy based on the consumer’s attribute set and call the chain code to perform attribute-based encryption on the symmetric key of the data. Next, it verifies whether the encrypted key and data CID are successfully recorded on the blockchain. The last step of the test is permission verification, simulating a user with the corresponding attribute set (a cardiology researcher at scientific research institution A) using his private key to successfully decrypt the key and finally obtain the original data, and at the same time verifies that a user without the corresponding attribute (such as Dept: ’Orthopedics’) fails to decrypt, to ensure the strictness of access control.Secure computing process: This test simulates a researcher (computation requester) uploading an encrypted data set containing anonymized health indicators of multiple patients and submitting a computing task. The test verifies that the third-party computing center can complete the task through homomorphic encryption computing without accessing the plaintext data and return the encrypted computing results. Finally, it verifies that the original requester can successfully decrypt and obtain the correct statistical results.

The test results show that this prototype system realizes all the core functions proposed in the paper, including user management, secure storage, data transaction sharing and data circulation calculation, verifying the correctness of the system design.

### 6.2. Performance Testing

The performance test aims to evaluate the network performance of the prototype system when processing chaincode operations. This section uses the Hyperledger Caliper tool to benchmark the two core operations of putValue (write) and getValue (read). At the same time, in order to quantify the performance overhead of cryptographic operations in this scheme, a baseline scheme (Baseline) that only performs simple data writing is also tested for comparison. All tests are performed in the experimental environment described in Section 6.1.

The detailed results of the performance test are summarized in Table 4. First, in terms of write operations, the throughput of the baseline solution can reach 124.3 TPS, while the throughput of the putValue operation in this solution, which contains complex cryptographic logic, is 64.5 TPS. The performance difference is mainly due to the fact that the chain code performs computationally intensive attribute-based encryption and other operations during the transaction endorsement phase, resulting in a significant increase in the average transaction latency from 0.11 s in the baseline to 0.59 s. It can be considered that this performance overhead is a reasonable trade-off in exchange for key security features such as data confidentiality and fine-grained access control. Secondly, the read operation (getValue) of this solution shows extremely high performance, with a throughput of up to 515.3 TPS and an average latency of only 0.01 s. This is in line with the architectural characteristics of Hyperledger Fabric, because the read operation as a query does not need to go through the full consensus process and can quickly obtain data directly from a single peer node.

To further explore the system’s resource overhead, Table 5 records the CPU and memory consumption of the core peer nodes during the write stress test. The results show that the average CPU usage of the nodes is maintained at around 26%, with a peak value of no more than 46%, and the average memory consumption is between 110–140 MB. These data show that when this solution achieves the performance level described, the resource utilization of its core components is at a healthy level that is far from saturation, which further verifies the stability and feasibility of the solution in actual deployment.

The fully homomorphic encryption encryption and decryption operations in the data transaction circulation module change with the polynomial modulus poly_modulus_degree. Increasing the polynomial modulus degree will improve the security of the encryption scheme because it increases the difficulty for attackers to crack the encryption, but at the same time the complexity increases, resulting in longer processing time. Figure 3 shows the time of encryption and decryption operations under different polynomial moduli.

In this solution, the CKKS parameters are set to polynomial modulus 8192, scaling factor $2^{40}$, and security level 128-bit. The sample sizes of the calculated data are set to 1000, 1500, 2000, 2500, 3000, and 3500, respectively. Multiple operations are performed, and the average time of each calculation task under each sample size is calculated. The results are shown in the Figure 4.

According to the chart, the average time required for the third-party computing center to perform homomorphic computing can be obtained. It changes according to the calculation sample size and calculation complexity, and generally meets the system requirements.

In summary, the test results of this prototype system verify the feasibility and effectiveness of the privacy protection model of data transaction circulation based on blockchain for the Internet of Things (IoT). Experiments have shown that this solution can provide solid technical support for the secure and efficient transaction and circulation of massive and heterogeneous IoT data, and also provide a valuable practical case for the application of blockchain technology in the key field of IoT data privacy protection.

## 7. Conclusions and Future Work

In view of the severe challenges faced by data transactions in the Internet of Things environment, such as lack of trust, privacy leakage and limited device resources, this paper designs and implements a secure data transaction and privacy protection framework based on blockchain. Through an innovative multi-layer security architecture, the framework organically integrates attribute-based encryption (ABE) to achieve dynamic fine-grained access control, game-theoretic pricing model to balance the interests of multiple parties, and fully homomorphic encryption (CKKS) to support secure computing of encrypted data. The functional and performance test results of the prototype system verify the correctness and feasibility of the scheme in architectural design, and provide a complete technical solution for achieving secure and efficient data circulation in a decentralized environment.

Although the scheme has shown feasibility in theory and experiments, it still has some limitations before large-scale deployment. First, performance overhead is the main challenge, and computationally intensive cryptographic operations may become a bottleneck when facing massive devices. Second, this research primarily validates security at a theoretical level. While we conducted formal security analysis based on cryptographic primitives, we lack experimental stress testing under simulated attack scenarios. For example, the system’s performance in the face of real-world threats such as data contamination attacks, key leakage, and user collusion remains to be further verified through experimental testing. Finally, the universality of the current pricing model is limited, and the complex and changeable value assessment factors in the real world need to be further modeled.

Future research will focus on the following directions: First, explore lightweight cryptographic schemes (such as zero-knowledge proofs or trusted execution environments) to optimize performance. Second, study the combination with decentralized identity (DID) to achieve more flexible and user-autonomous dynamic access control. Third, and most importantly, deeply integrate this data transaction framework with federated learning to design an end-to-end closed-loop solution from “trusted data circulation” to “privacy-enhanced data utilization”, which will be a very promising research direction.

## Figures and Tables

**Figure 1 sensors-25-04854-f001:**
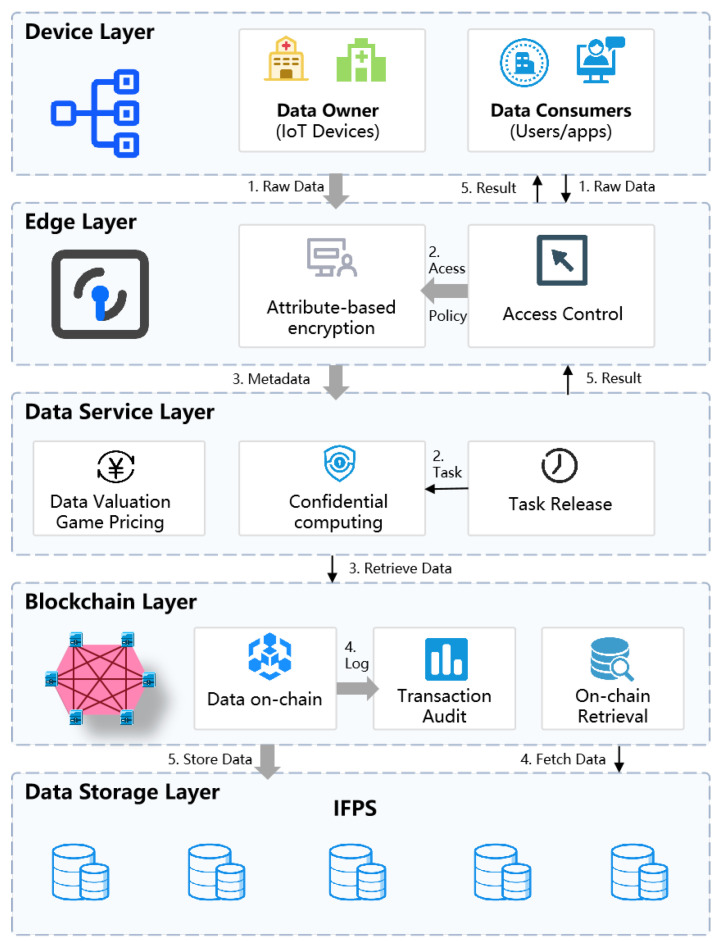
Layered architecture of data transaction and privacy protection framework based on blockchain.

**Figure 2 sensors-25-04854-f002:**
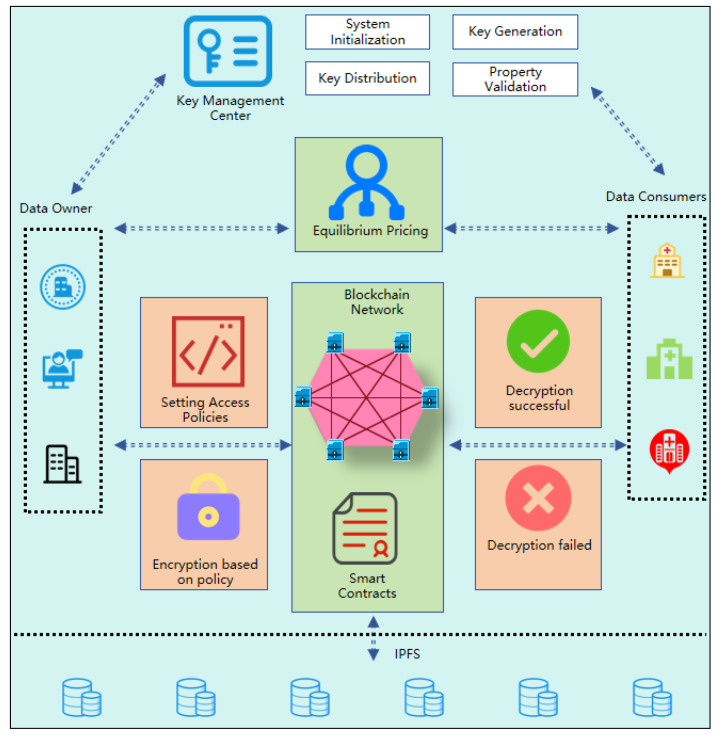
Overall architecture of data transaction sharing mechanism.

**Figure 3 sensors-25-04854-f003:**
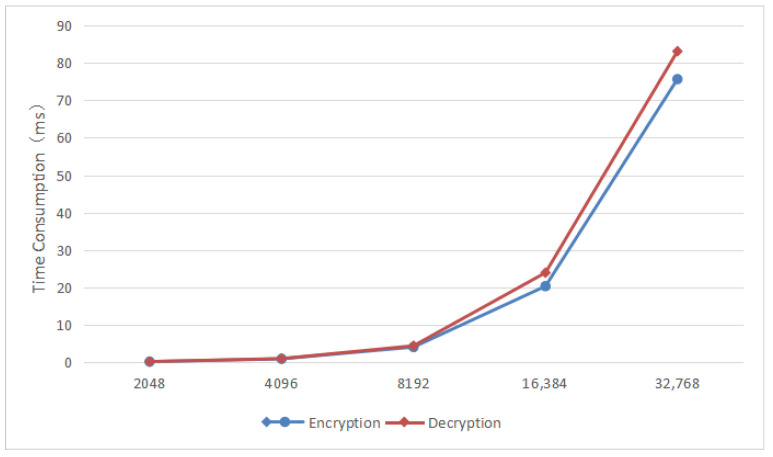
Encryption and decryption with the change of polynomial modulus.

**Figure 4 sensors-25-04854-f004:**
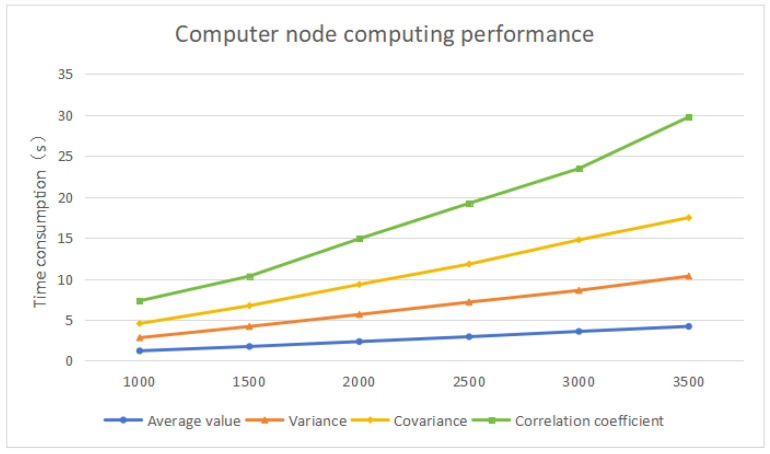
Comparison of computing task performance.

**Table 1 sensors-25-04854-t001:** Comparison of our scheme with other literature.

Reference	High-Availability Data Storage	Fine-Grained Access Control	Homomorphic Computation	Scalability
[11]	×	✓	×	×
[12]	✓	×	×	✓
[13]	✓	×	×	×
[14]	✓	×	×	✓
[9]	✓	✓	×	✓
[15]	×	×	✓	×
[28]	×	×	×	✓
[29]	✓	×	✓	✓
This Paper	✓	✓	✓	✓

**Table 2 sensors-25-04854-t002:** Classification of homomorphic encryption schemes.

Type	Properties	Schemes
Partially Homomorphic	Supports only one type of operation,	[40,41]
Encryption (PHE)	either addition or multiplication.	
Somewhat Homomorphic	Supports a limited number of mixed	[42]
Encryption (SHE)	addition and multiplication operations.	
Fully Homomorphic	Supports an unlimited number of	[43]
Encryption (FHE)	mixed operations.	

**Table 3 sensors-25-04854-t003:** System functional test results.

Functionality	Expected Result	Test Result
Fabric Node Registration	A new node obtains its certificate and key after registration.	✓
IPFS Data Upload/Download	Files can be uploaded to and downloaded from IPFS.	✓
Fabric On-chain/Query	Data can be uploaded to Fabric and queried from the chain.	✓
Data Owner Uploads Data	The data owner uploads files and related information as prompted, and the file enters the pending transaction list.	✓
System Negotiates Transaction	The system determines the optimal price based on data valuation and bids, and the transaction proceeds after confirmation from both parties.	✓
Data Owner Encrypted Sharing	Data access policy is set and data is encrypted; data information is recorded on-chain.	✓
Data Consumer Decrypts Data	Data is decrypted based on the user’s own attributes.	✓
Data File Encrypted Upload	The computation requester uploads a data file and specifies the computation task, after which the data is stored in encrypted form.	✓
Third-party Homomorphic Computation	The computation center retrieves the computation task from the chain and performs homomorphic computation.	✓
Data Decryption and Recovery	The computation requester retrieves the computation result from the chain and decrypts it.	✓

**Table 4 sensors-25-04854-t004:** Caliper’s performance test results on blockchain.

Name	Succ/Fail	Send Rate (TPS)	Max Latency (s)	Min Latency (s)	Avg Latency (s)	Throughput (TPS)
putValue	654/0	64.6	1.84	0.12	0.59	64.5
getValue	3741/0	515.3	0.05	0.00	0.01	515.3
Baseline	3728/0	125.1	0.34	0.06	0.11	124.3

**Table 5 sensors-25-04854-t005:** Resource consumption of peer nodes during the write benchmark.

Name	CPU% (max)	CPU% (avg)	Memory (max) [MB]	Memory (avg) [MB]
peer0.org2.example.com	44.30	26.20	111	109
peer0.org1.example.com	45.84	26.81	141	139

## Data Availability

The original contributions presented in the study are included in the article, and further inquiries can be directed to the corresponding author.

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
