# Peer review of "A Blockchain-Based Secure Data Transaction and Privacy Preservation Scheme in IoT System"

_sensors, 2025, doi:10.3390/s25154854_

Round 1
Reviewer 1 Report
Comments and Suggestions for Authors
Paper proposes a blockchain-based data transaction and privacy protection framework for IoT based on attribute-based encryption and a game theory-driven dynamic pricing model. To ensure confidential analysis of IoT data, authors proposed a secure computing scheme based on CKKS fully homomorphic encryption. The results are interesting and new; however, the presentation has several drawbacks that require further analysis, which should be done during the major revision stage.
1. Please mention the centralized data management model in more detail. Note that there exist privacy-preserving techniques that can even ensure information-theoretic security. Please extend the corresponding section and consider adding the following references:
[R1] doi.org/10.1561/0100000103
[R2] doi.org/10.1109/TIFS.2024.3450288
2. It would be great if authors could note the existence of the federated learning framework and its application in IoT settings:
[R3] doi.org/10.1145/3709023.3737688
[R4] doi.org/10.1145/3378679.3394528
3. Note that coded distributed computing and federated learning mentioned in the previous bullet points can be information-theoretically secure and do not employ computationally heavy homomorphic encryption. Could you please elaborate more on this? How would your scheme perform if homomorphic encryption were replaced?
4. Please provide a formal definition of the security model. Please consider proving security and privacy through the security experiment notion.
5. Please consider adding throughput comparison with state-of-the-art blockchain frameworks. Please give more details of the employed blockchain framework.
6. Is it possible to make the solution open-source?
Author Response
Comments 1: [Please mention the centralized data management model in more detail. Note that there exist privacy-preserving techniques that can even ensure information-theoretic security. Please extend the corresponding section and consider adding the following references:]
Response 1: [We sincerely thank the reviewers for their insightful comments. We fully agree that a more in-depth discussion of the centralized model and a comparison with the most advanced privacy-preserving techniques in the field are crucial to demonstrating the necessity and innovation of our work. Following your suggestions, we have revised the related work section. The specific changes are as follows: We elaborate on the specific challenges faced by the traditional centralized data management model in the IoT scenario, including single point failure risks, separation of data control rights, and risks of data tampering and abuse.
We acknowledge that there are also advanced privacy-preserving techniques in the centralized framework and have carefully studied the references you recommended. We have integrated the discussion on coded computation [R1] and verifiable coded computation [R2] into our argument to show a more comprehensive academic perspective. On this basis, we further clearly demonstrate the uniqueness and superiority of our proposed blockchain-based solution. (Lines 90-100)]
Comments 2: [It would be great if authors could note the existence of the federated learning framework and its application in IoT settings:
[R3] doi.org/10.1145/3709023.3737688
[R4] doi.org/10.1145/3378679.3394528]
Response 2: [Thank you for your insightful suggestions. We agree that discussing federated learning (FL) is important to clarify the scope and contribution of our work. In response, we have added new paragraphs (171-180) at the end of Section 2.3 (Dataflow Computation Schemes).]
Comments 3: [Note that coded distributed computing and federated learning mentioned in the previous bullet points can be information-theoretically secure and do not employ computationally heavy homomorphic encryption. Could you please elaborate more on this? How would your scheme perform if homomorphic encryption were replaced?]
Response 3: [Thank you for raising this insightful and important question. We fully agree that federated learning and encoded computation are state-of-the-art techniques that can provide strong privacy protection at a lower cost. To respond to your concerns and clarify our design intentions, we have added a new paragraph at the end of Section 4.3.1 to elaborate on this. In this paragraph, we first acknowledge the computational overhead challenges of fully homomorphic encryption. Then, we defend our technical choices from two key perspectives. First, our scheme aims to support third parties to perform general statistical analysis on encrypted raw data, which is fundamentally different from the "cooperative modeling" goal of federated learning. Second, we emphasize the unique ability of the CKKS scheme in processing encrypted floating-point numbers, a flexibility that is difficult to replace with other technologies. Finally, we directly answer the question of "what if homomorphic encryption is replaced?" and conclude that this will make the core feature of our scheme "secure computation" impossible to achieve. We believe that the current computational overhead is a necessary trade-off to obtain this high level of computational flexibility and end-to-end privacy.]
Comments 4: [Please provide a formal definition of the security model. Please consider proving security and privacy through the security experiment notion.]
Response 4: [Thank you very much for this crucial suggestion. We fully agree, and to incorporate your suggestion, we have completely restructured and expanded Chapter 5 "Security Analysis" to include a more complete proof framework based on the concept of "security experiment". The specific changes are as follows:
Formal Definition of Threat Model and Security Goals (Section 5.1): We have added a new section "5.1 Threat Model and Security Goals". In this section, we first formally define the entities in the system, then clarify the threat model and elaborate on the attacker's capabilities. Finally, we list the two core security goals that this solution aims to achieve.
Argumentation Based on Security Experiments (Section 5.2): We have added a new section "5.2 Security Proof" and conducted a rigorous analysis of each security goal under this framework.]
Comments 5: [Please consider adding throughput comparison with state-of-the-art blockchain frameworks. Please give more details of the employed blockchain framework.]
Response 5: [Thank you very much for the reviewer's valuable suggestions on experimental evaluation. We fully agree to provide more detailed framework information and add performance comparison with the baseline solution. In order to adopt your suggestions, we have modified and supplemented Chapter 6 "Experiments". Added detailed information on the framework: We described in detail the topology of the Hyperledger Fabric v2.2.1 network used, including its composition of two organizations, two peers in each organization, and an ordering service using Raft consensus. Added a new throughput comparison test: We rewrote Section 6.2 "Performance Test" and replaced the original figure with a new comprehensive performance comparison table (Table 4). In the new test, we introduced a "baseline solution" that runs a chaincode that only performs simple data writes in the same network environment. At the same time, considering clarity and simplicity, the original Figure 4 was also changed to Table 5.]
Comments 6: [Is it possible to make the solution open-source?]
Response 6: [Thanks to the reviewer for the suggestion on open source. We strongly agree with the value of open source. Since our prototype system code is coupled with some internal projects, it is difficult to open source it completely. However, we are very happy to provide the implementation code of the core algorithm or more detailed implementation details to our peers under reasonable academic research requests to support the reproduction of related research.]
Reviewer 2 Report
Comments and Suggestions for Authors
Title: A Blockchain-Based Secure Data Transaction and Privacy Preservation Scheme in IoT System
General Overview
The authors propose a novel blockchain-based solution aimed at addressing secure data transactions and privacy preservation within Internet of Things (IoT) systems. The initiative to leverage blockchain for these critical challenges in IoT is timely and relevant.
Observations and Recommendations
Abstract
- "Prototype system testing verifies the functional completeness and performance feasibility of the scheme, providing a complete and effective technical solution to solve the trust, privacy and access control problems in IoT data sharing."
- Observation: The authors assert that their solution addresses "trust" issues. In the context of distributed systems, "trust" is a complex concept often referring to the reliance on entities or mechanisms within a system. The abstract uses "trust" without a clear definition or a precise articulation of how the proposed solution establishes or manages trust in a distributed environment. Blockchain inherently offers trustlessness (reducing the need for trusted third parties) rather than defining trust in a traditional sense.
- Recommendation: We kindly suggest updating the abstract to clarify the specific aspect of "trust" being addressed (e.g., establishing verifiable transactions, ensuring data integrity without a central authority, or mitigating reliance on single points of trust) or rephrasing to align with the distributed systems understanding of trust.
- Use Case Specificity:
- Recommendation: The abstract should explicitly state that the proposed solution is designed with healthcare use cases in mind, as this specificity is crucial for readers to understand the context and applicability of the work.
Introduction
- "In this context, privacy computing technology has attracted widespread attention as a solution that can make data 'available but invisible'."
- Recommendation: A relevant reference should be added to support this statement regarding privacy-preserving computation.
- "The traditional data circulation model"
- Recommendation: A reference to the traditional data circulation models being discussed should be included for academic rigor.
- "blockchain has also raised concerns about privacy protection."
- Recommendation: Specific references highlighting privacy concerns in blockchain technology should be provided.
- Paper Outline:
- Recommendation: The introduction should conclude with a clear outline of the paper's structure, guiding the reader through the subsequent sections.
Related Work
- Citation Format Consistency:
- Recommendation: For consistency and adherence to journal standards, we advise the authors to use only one citation format throughout the paper. The [XXXX] reference style, if chosen, should be applied uniformly.
- Paragraph Structure for New Ideas:
- Recommendation: When introducing and motivating a new idea or a distinct related work, it should be presented as a new paragraph to improve readability and logical flow.
- Table/Figures/Algorithms/Ecuations Reference and Placement:
- Recommendation: Authors should explicitly reference all visual elements in the manuscript (tables/figures/algorithms/ecuations) within the text. Furthermore, this Table 1, which likely emphasizes the advantages of the proposed solution over existing ones, would be more impactful if placed at the end of the "Related Work" section. This placement would effectively summarize the limitations of prior art and set the stage for the proposed solution.
Preliminaries
- Section Length and Content:
- Observation: This section appears to be excessively long, containing an extensive amount of detail on existing solutions and algorithms.
- Recommendation: A lighter version of this content, focusing only on the most essential background information directly relevant to understanding the proposed scheme, should be included. More detailed descriptions of existing solutions and algorithms are better suited for the "Related Work" section, where they can be discussed in the context of their contributions and limitations.
System Structure and Design
- Figure Referencing and Captions:
- Recommendation: All figures should be referenced in the text. Each figure caption should conclude with a period (.).
- Figure 1 and Layered Model Description:
- Observation: The statement "The model consists of the device layer, edge layer, data service layer, blockchain layer, and data storage layer from bottom to top. Each layer has its own responsibilities and jointly ensures the secure circulation and privacy protection of massive IoT data" is not adequately reflected in Figure 1. The figure lacks the detailed representation of these layers and their interactions.
- Recommendation: The authors should revise Figure 1 to accurately depict the described layered architecture. Crucially, they should detail the specific functionalities of each layer and explicitly describe the interactions and links between these layers to ensure clarity for the reader.
- Algorithm Referencing and Details:
- Recommendation: All algorithms should be explicitly referenced in the text. The authors should provide detailed explanations of the mechanisms used for random number generation within their algorithms. While the four algorithms appear valid and consistent with standard pairing-based Attribute-Based Encryption (ABE) schemes, their specific implementation details, especially concerning randomness, need to be fully articulated.
- Equation Referencing:
- Recommendation: All equations presented in the paper should be referenced in the text.
- IPFS Usage and Value Proposition:
- Recommendation: The authors should provide a detailed explanation of how IPFS (InterPlanetary File System) is utilized within their proposed solution. More importantly, they need to clearly articulate how this specific approach to data storage adds value compared to other existing storage systems, justifying its selection.
Experiments
- Methodology for Functional Testing:
- Recommendation: Authors should clearly explain the methodology employed to obtain the functional test results. They should also elaborate on the relevance of these tests within the context of the proposed use case (healthcare, as suggested). A detailed description of the overall testing methodology used is essential for reproducibility and credibility.
- Figure Quality and Replacement:
- Observation: Figures 3 and 4 are of very poor quality, making it difficult to extract relevant information.
- Recommendation: These figures should be replaced with tables. A tabular format would be much more effective in transmitting the relevant data clearly and concisely.
Conclusion (of the paper)
- Lack of Discussion Section:
- Observation: The paper lacks a dedicated discussion section.
- Recommendation: The authors should include a discussion section where they thoroughly argue the limitations of the proposed solution. This section should also emphasize how the solution specifically helps in the proposed use case (healthcare) and discuss potential future research directions.
Overall Conclusion
While the manuscript presents an interesting and relevant contribution to the field of secure data transactions and privacy preservation in IoT systems, it requires significant revisions before it can meet the standards for publication in a scientific journal. The observations and recommendations outlined above highlight areas that need substantial improvement in terms of clarity, academic rigor, detailed explanations, and presentation quality. Given its current form, the paper is not ready for publication. We strongly recommend the authors implement all requested observations and modifications to increase the value and scientific merit of the paper. Following these revisions, the paper could be considered for acceptance with modifications.
Author Response
Abstract
Comments 1: [We kindly suggest updating the abstract to clarify the specific aspect of "trust" being addressed (e.g., establishing verifiable transactions, ensuring data integrity without a central authority, or mitigating reliance on single points of trust) or rephrasing to align with the distributed systems understanding of trust.]
Response 1: [Thank you very much for this insightful comment from the reviewer. We agree that the word “trust” used in the Abstract is indeed imprecise. To address this, we have revised the Abstract to be more specific about how our solution enhances the trustworthiness of the system. Instead of using the general term “trust,” we explicitly state that our framework ensures “data integrity and verifiable transactions while alleviating reliance on trusted centralized institutions.” This is more in line with the core idea of “trustlessness” in distributed systems. The relevant changes have been highlighted on page 1, lines 21-24 of the revised manuscript.]
Comments 2: [The abstract should explicitly state that the proposed solution is designed with healthcare use cases in mind, as this specificity is crucial for readers to understand the context and applicability of the work.]
Response 2: [Thank you for your valuable suggestions. We fully agree that clarifying the application scenario is crucial for readers to understand the article. We have revised the abstract according to your suggestions, clearly stating that our proposed framework is specifically designed for the Internet of Medical Things (IoMT) scenario. The relevant changes have been highlighted in lines 8-9 on the first page of the revised manuscript.]
Introduction
Comments 1: [Recommendation: A relevant reference should be added to support this statement regarding privacy-preserving computation.
"The traditional data circulation model"
Recommendation: A reference to the traditional data circulation models being discussed should be included for academic rigor.
"blockchain has also raised concerns about privacy protection."
Recommendation: Specific references highlighting privacy concerns in blockchain technology should be provided.]
Response 1: [Thank you to the reviewer for pointing out the omissions in our citations. We agree that these assertions need to be supported by references to enhance academic rigor. We have added references to these three assertions as requested:
Added a reference to "Privacy computing technology" on page 2, line 42.
Added a reference to "Traditional data circulation model" on page 2, line 47.
Added a reference to "Privacy issues of blockchain" on page 2, line 55.
All changes have been highlighted in the revised manuscript.]
Comments 2: [ The introduction should conclude with a clear outline of the paper's structure, guiding the reader through the subsequent sections.]
Response 2: [Thank you for the reviewer's suggestion. We agree that a clear structural outline is important to guide the reader. We have added a new paragraph at the end of the introduction to summarize the main contents of the subsequent chapters of the paper. The relevant changes have been highlighted in the revised manuscript.]
Related Work
Comments 1: [For consistency and adherence to journal standards, we advise the authors to use only one citation format throughout the paper. The [XXXX] reference style, if chosen, should be applied uniformly.
Paragraph Structure for New Ideas:]
Response 1: [Thank you for the careful reminder from the reviewer. We agree that maintaining consistency in citation format is essential for the professionalism of the paper. We have carefully reviewed and revised the entire manuscript to ensure that all reference citation formats (e.g., multiple references are combined into [3,4,7]) are consistent throughout the paper.]
Comments 2: [When introducing and motivating a new idea or a distinct related work, it should be presented as a new paragraph to improve readability and logical flow.]
Response 2: [Thank you for your constructive comments on improving the structure of the manuscript. We agree that proper paragraph division is key to improving readability. We have carefully revised the entire manuscript, especially the Introduction and Related Work sections. Long paragraphs that discuss multiple different ideas or research works have now been broken up into shorter, more focused paragraphs.]
Comments 3: [Authors should explicitly reference all visual elements in the manuscript (tables/figures/algorithms/ecuations) within the text. Furthermore, this Table 1, which likely emphasizes the advantages of the proposed solution over existing ones, would be more impactful if placed at the end of the "Related Work" section. This placement would effectively summarize the limitations of prior art and set the stage for the proposed solution.]
Response 3: [Thank you for pointing out this important point about academic writing standards. We have carefully reviewed the entire paper and ensured that every figure, table, and algorithm is now clearly referenced in the main text, near where its related content is discussed. In addition, we agree that placing Table 1 at the end of the entire "Related Work" section can provide a more logical and powerful summary of the literature review. We have now moved Table 1 to the end.]
Preliminaries
Comments 1: [A lighter version of this content, focusing only on the most essential background information directly relevant to understanding the proposed scheme, should be included. More detailed descriptions of existing solutions and algorithms are better suited for the "Related Work" section, where they can be discussed in the context of their contributions and limitations.]
Response 1: [Thank you for your valuable suggestions. We agree that Chapter 3 "Background Knowledge" does contain too many unnecessary details, which affects the reading fluency of the article. Based on your comments, we have greatly streamlined this chapter. We removed the lengthy mathematical description of the CKKS homomorphic encryption scheme and only kept the parts that are necessary for understanding the subsequent content.]
System Structure and Design
Comments 1: [All figures should be referenced in the text. Each figure caption should conclude with a period (.).]
Response 1: [Thank you for pointing out these important details about formatting conventions. We have carefully revised the manuscript to address both points. We have ensured that each figure is now clearly referenced in the text. In addition, we have reviewed all figure captions and added a period at the end of each to ensure consistency in formatting and compliance with publication standards.]
Comments 2: [The authors should revise Figure 1 to accurately depict the described layered architecture. Crucially, they should detail the specific functionalities of each layer and explicitly describe the interactions and links between these layers to ensure clarity for the reader.]
Response 2: [Thank you for pointing out the important issue of inconsistency between the figure and the text. We fully agree that the original Figure 1 does not accurately reflect the five-layer architecture described in the article. Based on your valuable suggestions, we have completely redrawn Figure 1. The new Figure 1 now clearly shows the vertical layered structure of "device layer, edge layer, data service layer, blockchain layer, and data storage layer" from bottom to top.]
Comments 3: [All algorithms should be explicitly referenced in the text. The authors should provide detailed explanations of the mechanisms used for random number generation within their algorithms. While the four algorithms appear valid and consistent with standard pairing-based Attribute-Based Encryption (ABE) schemes, their specific implementation details, especially concerning randomness, need to be fully articulated.
Equation Referencing:]
Response 3: [We would like to thank the reviewers for their rigorous review and valuable suggestions on the details of our algorithm. We deeply realize that a clear and detailed description of the source of randomness is crucial to the rigor of a security scheme. We have made the following two changes: First, we have carefully checked the text and explicitly added references to Algorithms A1 to A4 in Appendix A in the corresponding locations describing the system initialization, key generation, encryption, and decryption processes in Section 4.2.2. And we have added a paragraph entitled "Randomness Assumptions" at the beginning of Chapter 5, which elaborates on the random number generation mechanism in detail.]
Comments 4: [All equations presented in the paper should be referenced in the text.]
Response 4: [Thank you for your careful reminder. We agree that clear references to all formulas are a basic requirement to ensure the rigor of the paper. We have carefully read the entire paper and numbered each formula in a single line, and added clear references to the text in the relevant context.]
Comments 5: [The authors should provide a detailed explanation of how IPFS (InterPlanetary File System) is utilized within their proposed solution. More importantly, they need to clearly articulate how this specific approach to data storage adds value compared to other existing storage systems, justifying its selection.]
Response 5: [Thank you for the reviewers' valuable comments on the IPFS section. We agree that a more detailed explanation of the use mechanism and core value of IPFS is essential to justify our technology selection. We have added a new paragraph after the general introduction of the five-layer architecture in Section 4.1. This paragraph is at the end of Section 4.1 and elaborates on the following two aspects: clarifying the synergy mechanism of "blockchain + IPFS" and discussing the core value of choosing IPFS.]
Experiments
Comments 1: [Authors should clearly explain the methodology employed to obtain the functional test results. They should also elaborate on the relevance of these tests within the context of the proposed use case (healthcare, as suggested). A detailed description of the overall testing methodology used is essential for reproducibility and credibility.
Figure Quality and Replacement:]
Response 1: [Thank you for your valuable suggestions. In order to enhance the credibility and reproducibility of the experiment, we have added detailed text descriptions to the original test result table (Table 3) in Section 6.1 "Functional Testing". We not only elaborated on how to verify each core function listed in the table one by one through simulation and scripted operations, but also explained in depth why these test links (such as node registration, data transactions, secure computing, etc.) are crucial to our core medical Internet of Things (IoMT) application scenarios, ensuring that the test methods are closely integrated with the application background.]
Comments 2: [These figures should be replaced with tables. A tabular format would be much more effective in transmitting the relevant data clearly and concisely.]
Response 2: [Thank you for the reviewer's suggestion. We fully agree that using tables can present the performance test data more clearly and professionally. According to your comments, we have removed the poor quality Figures 3 and 4 in the original manuscript and replaced them with new performance comparison tables (Tables 4 and 5). The new tables not only integrate the core data of the original two figures, but also add comparisons with the benchmark solution.]
Conclusion (of the paper)
Comments 1: [The authors should include a discussion section where they thoroughly argue the limitations of the proposed solution. This section should also emphasize how the solution specifically helps in the proposed use case (healthcare) and discuss potential future research directions.]
Response 1: [Thank you for your valuable suggestions on the conclusion of the paper. We fully agree, but considering the length of the paper, we have completely restructured Chapter 7 based on your comments, expanding it from a simple conclusion to a more in-depth "Conclusion and Outlook" chapter.]
Reviewer 3 Report
Comments and Suggestions for Authors
Although the manuscript is high quality, I have some recommendations to enhance its presentation, clarity, and accessibility:
- I suggest incorporating a succinct problem statement at the conclusion of the introduction to explicitly delineate the research gap and the originality of the suggested methodology.
- While the linked work section has important references, the authors might consider adding more recent studies (2023–2025) about blockchain and privacy-preserving methods in IoT systems to better relate their work to the latest developments.
- The technical descriptions of CKKS and CP-ABE are thorough. Nevertheless, including the complete algorithm listings (Algorithms 1–4) in an appendix or additional material may improve readability and preserve the narrative continuity of the main text.
- It is advisable to refrain from positioning a subchapter title directly behind a chapter title. The authors should incorporate a concise introduction paragraph or contextual material following each chapter title to enhance reading and ensure a seamless transition into sub-sections.
- Certain portions are laden with intricate technical details and mathematics. Incorporating concise, intuitive explanations alongside formal definitions may enhance the paper's accessibility to a wider readership.
- Incorporating a column in Table 1 for "Scalability" or "IoT Suitability" could elucidate the practical benefits of the proposed strategy.
- A specific "Future Work" section could be added to look into possible improvements, like enabling real-time IoT data applications or working with federated learning frameworks.
Author Response
Comments 1: [I suggest incorporating a succinct problem statement at the conclusion of the introduction to explicitly delineate the research gap and the originality of the suggested methodology.]
Response 1: [Thank you very much for the reviewer's valuable suggestions. We agree that adding a clear problem statement at the end of the introduction can better guide readers and highlight the core goals of our work. Following your suggestion, we have added a concise problem statement at the end of the introduction (lines 68-74). This paragraph clearly summarizes the core research problem we aim to solve and its three key sub-problems: decentralized access control, dynamic pricing models, and secure computing on encrypted data.]
Comments 2: [While the linked work section has important references, the authors might consider adding more recent studies (2023–2025) about blockchain and privacy-preserving methods in IoT systems to better relate their work to the latest developments.]
Response 2: [Thank you for your valuable suggestions. We agree that incorporating the latest research is essential for positioning our work. In the revised manuscript, we updated the literature review in Section 2.1, replacing some older references with the latest research from 2023-2025 (specifically 8, 9, 10). As well as (5, 6, 25, 26) on federated learning, etc.]
Comments 3: [The technical descriptions of CKKS and CP-ABE are thorough. Nevertheless, including the complete algorithm listings (Algorithms 1–4) in an appendix or additional material may improve readability and preserve the narrative continuity of the main text.]
Response 3: [Thank you for your suggestion. We fully agree that moving the detailed algorithm list to the appendix can make the narrative logic of the main text more coherent. We have moved the complete contents of Algorithms 1 to 4 in Chapter 4 to a new appendix (Appendix A) at the end of the paper. We have retained the references and functional descriptions of these algorithms in the corresponding positions in the main text.]
Comments 4: [It is advisable to refrain from positioning a subchapter title directly behind a chapter title. The authors should incorporate a concise introduction paragraph or contextual material following each chapter title to enhance reading and ensure a seamless transition into sub-sections.]
Response 4: [Thank you for your valuable suggestion. We have carefully checked the entire text and added a concise introductory paragraph after each major chapter heading (Chapter 2, 3, and 4).]
Comments 5: [Certain portions are laden with intricate technical details and mathematics. Incorporating concise, intuitive explanations alongside formal definitions may enhance the paper's accessibility to a wider readership.]
Response 5: [Thank you for the reviewers' valuable suggestions. We fully agree that adding intuitive explanations next to complex mathematical and technical definitions is crucial to improving the readability of the paper. To address this issue, we carefully reviewed the entire paper and revised several sections where the most technical details are concentrated. After introducing the CKKS homomorphic encryption scheme in Section 3.2, we added an intuitive explanation, using metaphors such as "denoiser" to illustrate the role of core components such as relinearization keys. After introducing the CP-ABE scheme in Section 3.3, we used a metaphor of "a safe with multiple keyholes" to help readers intuitively understand the principles of its encryption and access control. In Section 4.2.3 on data integrity assessment, we added a short, non-mathematical explanation after each core mathematical formula (such as state estimation model, objective function, etc.) to explain the practical significance of the formula in medical scenarios.]
Comments 6: [Incorporating a column in Table 1 for "Scalability" or "IoT Suitability" could elucidate the practical benefits of the proposed strategy.]
Response 6: [Thank you for your excellent suggestions to better highlight the practical benefits of our approach. We agree that scalability is a key metric for IoT systems. We have modified Table 1 to include a new column called “Scalability”.]
Comments 7: [A specific "Future Work" section could be added to look into possible improvements, like enabling real-time IoT data applications or working with federated learning frameworks.]
Response 7: [We thank the reviewers for their constructive suggestions on the direction of "future work". We agree that it is important for this study to clearly plan future research directions, especially to explore the combination with cutting-edge technologies such as federated learning. We have completely restructured Chapter 7, expanding it from a simple conclusion to a more in-depth "Conclusion and Outlook" chapter.]
Round 2
Reviewer 1 Report
Comments and Suggestions for Authors
Accept as authors have addressed all comments
Author Response
Thank you very much for your recognition of our manuscript. We sincerely appreciate your valuable comments and professional guidance.